# ENCODING SPEAKER-SPECIFIC LATENT SPEECH FEATURE FOR SPEECH SYNTHESIS

## ABSTRACT

In this work, we propose a novel method for modeling numerous speakers, which enables expressing the overall characteristics of speakers in detail like a trained multi-speaker model without additional training on the target speaker's dataset. Although various works with similar purposes have been actively studied, their performance has not yet reached that of trained multi-speaker models due to their fundamental limitations. To overcome previous limitations, we propose effective methods for feature learning and representing target speakers' speech characteristics by discretizing the features and conditioning them to a speech synthesis model. Our method obtained a significantly higher similarity mean opinion score (SMOS) in subjective similarity evaluation than seen speakers of a best-performing multi-speaker model, even with unseen speakers. The proposed method also outperforms a zero-shot method by significant margins. Furthermore, our method shows remarkable performance in generating new artificial speakers. In addition, we demonstrate that the encoded latent features are sufficiently informative to reconstruct an original speaker's speech completely. It implies that our method can be used as a general methodology to encode and reconstruct speakers' characteristics in various tasks.

## 1 INTRODUCTION

Recently, research on modeling numerous speakers in the real world has been actively studied. Previous works (Gibiansky et al., 2017; Ping et al., 2018; Chen et al., 2020; Kim et al., 2020; 2021) used a trainable speaker embedding matrix to learn the speech characteristics of each speaker in one model to model multiple speakers effectively; this is commonly referred to as multi-speaker speech synthesis. Because the method enables a similar expression of each speaker's characteristics and the sharing of common information among speakers, it is effective in synthesizing the speech of multiple speakers in high quality with relatively less training data than training each speaker in one model. However, the model must be trained for all speakers whenever a new speaker is added, and synthesizing high-quality speech may not be possible for speakers with a relatively small dataset. Considering the above aspects, the modeling of a fairly large number of speakers by extending this method is limited. A fine-tuning approach was employed to mitigate the necessity and limitations of the training process (Chen et al., 2019; 2021; Huang et al., 2022). This method fine-tunes a sufficiently trained model using a small amount of target speaker data. However, it partially overcomes problems of the training process.

The most actively studied method to address the limitations of training approaches is zero-shot speech synthesis. The method synthesizes speech that represents speech characteristics similar to those of the corresponding speaker by conditioning a speaker vector obtained from the short audio of a target speaker. This is achieved by training a speaker encoder and speech synthesis model with large multi-speaker datasets (Jia et al., 2018; Hsu et al., 2019; Cooper et al., 2020; Casanova et al., 2022). A speaker vector is obtained from a speaker encoder with short audio of a target speaker as input, and the vector is conditioned to a speech synthesis model to express the target speaker's speech characteristics. This method seems to enable synthesizing speech audio similar to that of the target speaker for an infinite number of speakers without additional training. However, it has fundamental limitations in expressing a speaker's overall speech characteristics. Since humans express different timbres and prosody depending on given contents, modeling overall speech characteristics according to the given content is crucial to synthesizing natural speech like humans (Hayashi et al., 2019; Kenter et al., 2020; Xu et al., 2021; Jia et al., 2021; Tan et al., 2022). The timbre and prosody

expressed in speech audio are aligned with its content, and only a small portion of the speaker's speech characteristics are revealed in a short speech; therefore, the obtained vector from the short reference audio represents a tiny part of a speaker's speech characteristics that vary depending on given contents. Take the following two sentences as examples: "I'm so happy!" and "I'm so sad.". The speaker's psychological state uttering the two sentences will be greatly different, and it will be revealed as speech characteristics. If the sentence "I'm so sad." is synthesized with a recorded speech audio of "I'm so happy!" as the reference audio, the synthesized speech will be unnatural and show quite different speech characteristics from the actual speech when the target speaker utters "I'm so sad.". It will be synthesized with bright timbre and prosody contrasting with typical human speech.[1] As a result, depending on the given reference audio, sometimes natural and similar speech characteristics are expressed, but oppositely unnatural and clearly different speech characteristics are often expressed. For the same reason, it is also limited to imitating the speaker's particular pronunciations and accents, which can be differently expressed depending on the content; it shows quite different results from the ground truth and trained multi-speaker model. Previous works (Zhou et al., 2022; Yin et al., 2022; Choi et al., 2023) used multiple features related to input content; the methods do not condition speakers' overall speech characteristics in training text-to-speech models and are still focused on following short reference audio. Thus, the fundamental limitations remain.

Additionally, this method usually utilizes a speaker encoder trained for a speaker-verification task (Jia et al., 2018; Cooper et al., 2020; Casanova et al., 2022). Its typical objectives are to distance speakers to distinguish them easily. Therefore, speakers with similar timbre or speech characteristics can be learned to be excessively further apart than they are similar, and the learned space for speaker vectors can be sparse and discontinuous. (This is discussed in detail in Appendix A.1.) These features can make obtaining appropriate speaker vectors for speech synthesis from the learned space limited, and synthesis models tend to learn each vector as an independent condition rather than a space. It often yields a definite failure to express speech characteristics similar to the target speaker and to synthesize accurate speeches.

Recently, zero-shot speech synthesis with prompting mechanisms (Wang et al., 2023; Shen et al., 2023; Le et al., 2023) has shown good results. These works demonstrated an excellent ability to preserve the timbre and prosody of the prompt. According to its fundamental operating principle, this method strongly relies on a given prompt to preserve speech characteristics. The results of the works also confirm that the synthesized speeches follow the prompts rather than the context of the given content in expressing speech characteristics such as timbre and prosody. Considering the training principles, objectives, and evaluation methods, it is natural results since the method that most closely follows the given prompts will yield better performance in that manner. However, considering that humans express timbre and prosody quite differently depending on contents, this method significantly differs from human behavior. This method is advantageous when a prompt suitable for the content is given, but conversely, it is disadvantageous when any content can be given. To address the problem, a method to encode the overall speech features of speakers and express the features according to the given content appropriately is needed.

To address the problems of the previous methods, we propose a method for **E**ncoding speaker-specific **L**atent speech **F**eatures for speech synthesis (**ELF**). Our main goal is to present a method to express new speakers' overall speech characteristics according to given contents, like multi-speaker speech synthesis, without a training process. We first encode various speech features from speakers' speech into a dense and continuous distribution. Then, we cluster these speech features to obtain discretized representative points. By observing various human speeches, we can intuitively confirm that human speech characteristics cannot be discretized. Therefore, it is difficult to expect good results from using these discretized features individually. We took the inspiration that a weighted sum through attention is essentially a linear combination of vectors and enables sampling a point on a continuous space that is formed with the given vectors. We designed a module to fuse the discretized speech feature into a hidden representation of the content through attention. It enables not only the speech synthesis model to learn the speech feature space but also the features to be fused to express the given content naturally.

ELF shows better speaker similarity, stability, and equivalent naturalness without additional training compared to a multi-speaker model trained with the target speaker's dataset. Moreover, it surpasses a zero-shot model by a significant margin in the zero-shot scenario. In addition, ELF demonstrates

---

[1] Related samples are available in the second section of our demo: https://osgfqwqbsfcr.github.io/elf-demo/

superior speaker blending performance, which means that the latent space is well-formed according to speaker similarity. It allows the synthesis of high-quality speech with a newly generated artificial speaker. Furthermore, we show that ELF enables the complete reconstruction of a speaker's speech solely with the encoded speech features. This implies that the proposed latent representation is informative enough to express the entire target speaker's speech, even without directly inputting content. Additionally, we demonstrate that ELF has the ability to synthesize high-quality speech in a cross-lingual manner. We present the effectiveness of our method through subjective and objective evaluations.

## 2 METHOD

### 2.1 OVERVIEW

In order to enable the text-to-speech model to synthesize speech according to the overall speech characteristics of an unseen speaker and a given content, it is essential to condition speakers' overall speech characteristics to the text-to-speech model in training. Therefore, ELF consists of two stages. The first stage encodes the individual speech features of each speaker, and the second stage synthesizes speech that expresses the target speaker's speech characteristics through conditioning the encoded features. The overall structure of ELF is shown in Figures 1 and Figures 3. We describe the details in the following sections.

### 2.2 SPEECH FEATURE ENCODING

The speech synthesis task aims to synthesize a speech that expresses the speech characteristics of a target speaker similarly and naturally to the ground truth recorded by humans. Therefore, an important point in encoding speech features is obtaining representations capable of reconstructing the original speech in high quality. We take the inspiration that an autoencoder is a structure that encodes input data to latent representation and decodes it to reconstruct the original data; we design the speech feature encoding network (SFEN) as an autoencoder trained to reconstruct the raw waveform from the input mel-spectrogram through encoding and decoding. We adopt the generator that showed superior performance in reconstructing raw waveforms of the previous work (Kong et al., 2020) as the decoder in our model. We also introduce the adversarial learning mechanism and the discriminator to our work to increase reconstruction performance. Furthermore, we use a variational approach (Goodfellow et al., 2014) to fit the unit Gaussian prior to increasing the possibility of sampling the point in the learned space through a combination of latent representations.

Therefore, SFEN is a variational autoencoder; it is trained to maximize the variational lower bound, also called the evidence lower bound(ELBO), of the intractable marginal log-likelihood of data $\log p_\theta(x)$:

$$\log p_\theta(x) = \mathbb{E}_{q_\phi(z|x)}\left[\log \frac{p_\theta(x,z)}{q_\phi(z|x)}\right] + \mathbb{E}_{q_\phi(z|x)}\left[\log \frac{q_\phi(z|x)}{p_\theta(z|x)}\right]$$
$$\geq \mathbb{E}_{q_\phi(z|x)}\left[\log p_\theta(x|z)\right] - D_{KL}(q_\phi(z|x)||p(z)) \quad (1)$$

where $z$ is the latent variable generated from the prior distribution $p(z)$, unit Gaussian, $p_\theta(x|z)$ is the likelihood function of a data point $x$, and $q_\phi(z|x)$ is an approximate posterior distribution. The training loss is then the negative ELBO, which is the sum of reconstruction loss $-\log p_\theta(x|z)$ and KL divergence $D_{KL}(q_\phi(z|x)||p(z))$.

We define the reconstruction loss as the $L_1$ loss between the input mel-spectrogram $s$ and the mel-spectrogram $\hat{s}$ from the reconstructed waveform. This can be viewed as a maximum likelihood estimation assuming a Laplace distribution for the data distribution and ignoring constant terms.

To introduce adversarial learning to train SFEN, following the previous work (Kong et al., 2020), we adopt the discriminator and the loss functions, the least-squares loss function Mao et al. (2017) for adversarial training, and the feature-matching loss (Larsen et al., 2016). Then, the total loss for

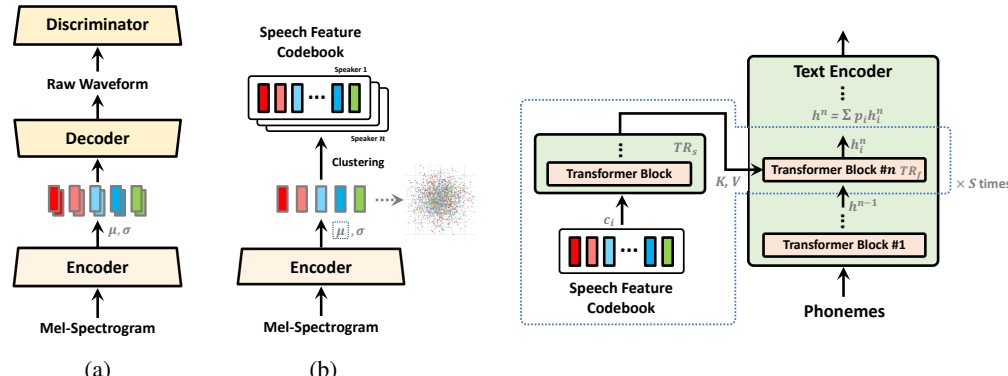

Figure 1: (a) Training procedure of SFEN. (b) Inference procedure of SFEN.

Figure 2: Prior encoder of speaker blending.

training SFEN and the discriminator is defined as

$$\mathcal{L}_{sf}(D_{sf}) = \mathbb{E}_{(y,s)}\Big[(D_{sf}(y) - 1)^2 + (D_{sf}(G_{sf}(s)))^2\Big], \tag{2}$$

$$\mathcal{L}_{sf}(G_{sf}) = \mathbb{E}_z\Big[(D_{sf}(G_{sf}(s)) - 1)^2\Big] + \mathbb{E}_{(y,s)}\Big[\sum_{l=1}^{T}\frac{1}{N_l}\|D_{sf}^l(y) - D_{sf}^l(G_{sf}(s))\|_1\Big]$$
$$+ \lambda_{sf}\|s - \hat{s}\|_1 + D_{KL}(q_\phi(z|s)\|p(z)) \tag{3}$$

where $D_{sf}$ and $G_{sf}$ denote the discriminator and SFEN, respectively, $y$ is the ground truth waveform, $T$ denotes the total number of layers in the discriminator, and $D_{sf}^l$ is the output feature map of the $l$-th layer of the discriminator with $N^l$ number of features. We set $\lambda_{sf} = 45$ following the previous work (Kong et al., 2020).

Speaker-specific speech features are obtained using the encoder of the learned autoencoder networks. First, we input the mel-spectrogram generated from the target speaker's audio to the encoder and obtain $\mu$ and $\sigma$, the distribution parameters, as the output, which has the same time-step as the mel-spectrogram. Next, we collect all the $\mu$ values of all the target speaker's audio and cluster the values using k-mean++ (Arthur & Vassilvitskii, 2007). Then, we use the centroids of the clusters as the latent speech feature codebook for the speaker. These features are a finite number of non-contiguous vectors; however, the vectors are combined when conditioned to a speech synthesis model, which leads to an effect similar to sampling a point in the continuous space.

## 2.3 TEXT-TO-SPEECH WITH SPEECH FEATURE CONDITION

The codebook obtained through the speech feature encoding process and clustering consists of a finite number of vectors. It is not reasonable to individually sample these vectors and use them as features because human speech characteristics cannot be discretized. Instead, we combine and add these features to an intermediate feature of a speech synthesis model with softmax attention scores, which allows an effect similar to sampling a speech feature point in continuous space. We apply this method to a text-to-speech (TTS) task based on the previous work (Kim et al., 2021) that showed superior performance. Various points in the model can be candidates to condition the speech features; we design the combined speech feature to be fused to the intermediate feature of the text encoder, considering some factors, such as a speaker's particular pronunciation and intonation, significantly influences the expression of each speaker's characteristics but are not presented in the input text. We first combine the vectors in the speech feature codebook using transformer blocks without positional information and add them to the intermediate feature of the text encoder through multi-head attention so that the factors can be effectively captured in modeling the prior distribution.

Since we condition the encoded speech features to the prior encoder, the prior distribution differs from the previous work (Kim et al., 2021). The conditional prior in our design is defined as

$$p_\theta(z|c_{text}, c_{sf}, A) \tag{4}$$

where $c_{text}$, $c_{sf}$, and $A$ denote the input phonemes, the speech feature codebook matrix of the target speaker, and the alignment between the phonemes and latent variables, respectively.

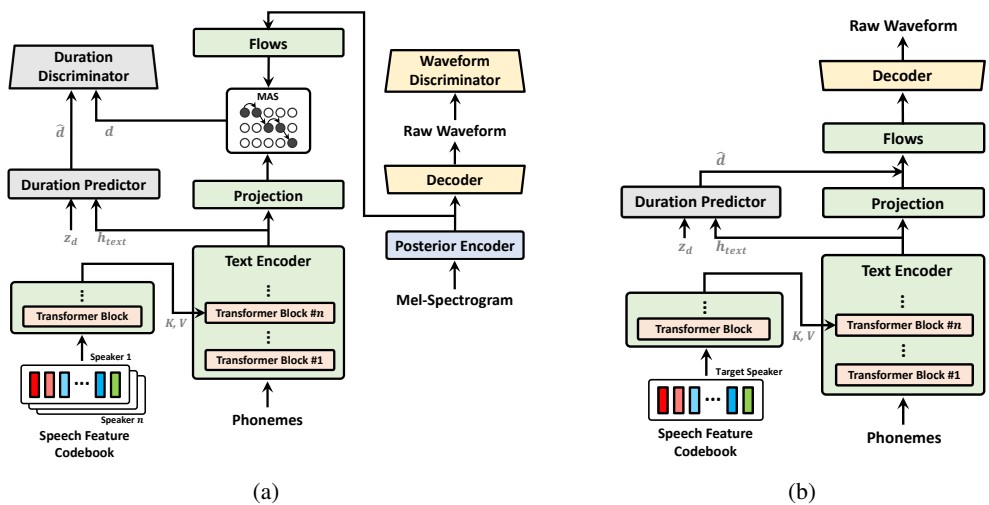

Figure 3: (a) Training procedure of TTS model. (b) Inference procedure of TTS model.

We change the duration predictor and normalizing flows of the previous work (Kim et al., 2021). As shown in the figure, we design a relatively simple duration predictor with adversarial learning, which models natural duration. We use the hidden representation of the text $h_{text}$, which is the output of the prior encoder and Gaussian noise $z_d$ as the input of the generator; the $h_{text}$ and duration obtained using monotonic alignment search(MAS) in the logarithmic scale denoted as $d$ or predicted from the duration predictor denoted as $\hat{d}$ are used as the input of the discriminator. We use two types of losses; the least-squares loss function (Mao et al., 2017) for adversarial learning and the mean squared error loss function. Then, the total loss function is defined as

$$\mathcal{L}_{dur}(D_{dur}) = \mathbb{E}_{(d,z_d,h_{text})}\Big[(D_{dur}(d, h_{text}) - 1)^2 + (D_{dur}(G_{dur}(z_d, h_{text}), h_{text}))^2\Big], \quad (5)$$

$$\mathcal{L}_{dur}(G_{dur}) = \mathbb{E}_{(z_d,h_{text})}\Big[(D_{dur}(G_{dur}(z_d, h_{text})) - 1)^2\Big] + \lambda_{dp}MSE(G_{dur}(z_d, h_{text}), d) \quad (6)$$

where $D_{dur}$ and $G_{dur}$ denote the discriminator and the duration predictor, respectively.

Also, capturing long-term dependencies can be crucial when transforming distribution because each part of the speech is related to other parts that are not adjacent. Therefore, we add a small transformer block with the residual connection into the normalizing flows to enable the capturing of long-term dependencies.

## 2.4 SPEAKER BLENDING

In Section 2.3, we described the method to obtain fused features to express a target speaker's characteristics with the speech feature codebook and the intermediate feature of the text encoder. In a similar manner, synthesizing speech in which the characteristics of multiple speakers coexist is possible through modifying the prior encoder to sum the intermediate features of multiple speakers at a specific ratio. The modified prior encoder is shown in Figure 2. Similar to TTS for an individual speaker, we first obtain the fused intermediate features through the attention with each speaker's speech features and the intermediate feature of the text encoder. Then, we multiplied the fused intermediate features with the weights according to the given proportions for each speaker and summed them all. Then, the final fused feature from multiple speakers is obtained as

$$h_i^n = TR_f(h^{n-1}, TR_s(c_i)), \qquad h^n = \sum_{i=1}^{S} p_i h_i^n \qquad (7)$$

where $S$ denotes the number of blending target speakers, $h^n$ is the output of $n$-th transformer block in the text encoder, $h_i^n$, $c_i$, and $p_i$ are $i$-th speaker's fused intermediate feature, codebook, and blending proportion, respectively, $TR_f$ and $TR_s$ are the transformer blocks to combine the vectors in the codebook and the transformer block to fuse the intermediate feature of the text encoder and the output of $TR_f$, respectively. In the multi-head attention in $TR_f$, $h^{n-1}$ is used as query, and the output of $TR_s$ is used as key $K$ and value $V$.

## 2.5 Speech Feature-to-Speech with Text Condition

We present a new method for synthesizing speech with high quality that expresses a target speaker's characteristics only with combined speech features. Since the vectors in a codebook are sampled from a latent space that can reconstruct a target speaker's speech, it can be possible to synthesize a target speaker's speech only with appropriately combined speech features if the learned latent space is sufficiently informative. Thus, we design a method to combine speech features to synthesize speech corresponding to a given text. We modify the prior encoder of the TTS model in Section 2.3, as shown in Appendix A.4. First, we obtain intermediate features from each transformer block with speech features and a phoneme sequence. Then, we compute attention scores with multi-head attention between the two intermediate features as query and key, respectively. Then, we use the weighted sum of the speech features on the attention scores as the input to the next module. Therefore, the intermediate feature from a phoneme sequence is used only for calculating the attention scores. This method can be easily extended to various tasks where information to combine the speech features is available.

## 3 Experiments

### 3.1 Datasets

Two public datasets were used to train SFEN, TTS model and the speech feature-to-speech model. We used the LibriTTS (Panayotov et al., 2015) dataset, which consists of audio recordings of 2,456 speakers for a total duration of approximately 585.80 hours. We split the dataset into the training(2,446 speakers, 583.33 hours) and test(10 speakers, 2.47 hours) sets. We used the VCTK (Veaux et al., 2017) dataset containing 43.8 hours of speech from 108 speakers with various speech characteristics. The dataset was split into the training(97 speakers, 39.46 hours) and test(11 speakers, 4.34 hours) sets. We followed the previous work (Casanova et al., 2022) to select the test speakers from both datasets. We trimmed the beginning and ending silences and downsampled to 22.05kHz for all audio clips.

### 3.2 Speech Feature Encoding

To train SFEN, we used the windowed training and the same segment size following HiFi-GAN (Kong et al., 2020). Each segment was converted into an 80-dimensional mel-spectrogram and used as the input. FFT, window, and hop size were set to 2048, 2048, and 1024, respectively. The encoder is composed of convolution blocks. The decoder structure was the same as that of the HiFi-GAN (Kong et al., 2020) V1 model. Because the settings of mel-spectrogram generation differ from HiFi-GAN (Kong et al., 2020), the upsampling kernel sizes and factors were adjusted to [16, 16, 8, 8] and [8, 8, 4, 4], respectively. The output latent representation of the encoder was divided into $\mu$ and $\sigma$ with 2048 dimensions for each frame. After training, the $\mu$ values of each speaker were clustered into 512 clusters using the k-means++ (Arthur & Vassilvitskii, 2007) algorithm. The latent speech feature codebook for each speaker is composed of the centroids of the clusters.
We describe the details of optimization settings in Appendix A.5

### 3.3 Text-to-Speech

For the TTS model, the clean subsets of the LibriTTS dataset and the VCTK dataset were used from the training dataset described in Section 3.1. We used the same test set described in Section 3.1. We used 80-dimensional mel-spectrogram to calculate the reconstruction loss. In contrast to the previous work (Kim et al., 2021), we used the same mel-spectrograms as the input of the posterior encoder. We converted text sequences into International Phonetic Alphabet sequences using open-source software (Bernard, 2021), and fed the text encoder with the sequences. The multi-head attention mechanism (Vaswani et al., 2017) was used between the text encoder's fifth layer output and the output of the transformer blocks for combining the speech feature in a codebook.
We describe the details of optimization settings and reusing parameters in Appendix A.6

### 3.4 SPEECH FEATURE-TO-SPEECH

To confirm that the proposed latent speech features are sufficiently informative, we conducted speech feature-to-speech experiments. The modified parts of the TTS model are described in Appendix A.4. The training method and parameters are the same as the TTS model, except that the speech feature-to-speech model is trained up to 800k steps with random initialization.

## 4 RESULTS

### 4.1 TEXT-TO-SPEECH

We chose the best-performing models for which official implementations were publicly available as the comparison models: VITS (Kim et al., 2021) for the training method and YourTTS (Casanova et al., 2022) for the zero-shot method. We used all speakers of the LibriTTS and VCTK datasets to train VITS. For an accurate comparison, YourTTS and the proposed model used the same datasets and the test speakers described in Section 3.1. We randomly sampled 500 (LibriTTS 250, VCTK 250) of the test speakers' data as the evaluation set. We sampled the evaluation set with a minimum text length of 30 characters and a maximum of 200 characters. The evaluation set was not included in training VITS.

For the audio conditioning to YourTTS, we matched each data in the evaluation set with one randomly sampled audio from the same speaker's data. Because the length of conditioned audio can be crucial to express speech characteristics in a zero-shot TTS model, we used a length constraint in the matching. Since audio in the VCTK dataset is relatively shorter than the LibriTTS dataset, using the same length constraint results in a shortage of candidates for the random matching in the VCTK dataset. Therefore, we used audio at least 3 and 5 seconds in VCTK and LibriTTS, respectively.

We conducted evaluations to confirm that the quality of synthesized speech varies according to the amount of data used in generating the speech feature codebook. We used three variations: one audio sample, 20 audio samples, and all available audio samples. For evaluation using one audio sample, we used the same matched audio as YourTTS. Because the total number of output features from the encoder with one input audio is shorter than the size of the codebook, we use the $\mu$ values as the speech feature condition without the clustering. In the evaluation using 20 audio samples, we matched each data in the evaluation set with randomly sampled 20 audio from the same speakers with the same length constraint. The results of the variations are shown in Table 1 in the manuscript as "# of audio=1", "# of audio=20", and "all audio", respectively.

We conducted two kinds of subjective evaluations: MOS for naturalness and SMOS for speaker similarity. We used a similar method in the previous work (Jia et al., 2018) for the SMOS evaluation. Because speaker characteristics, such as pronunciation and intonation, can vary depending on the content, the SMOS evaluations were set up so that the text of the reference audio and synthesized sample are identical. It allowed raters to assess the similarity accurately.

The scales of both MOS and SMOS ranged from 1 to 5 with 1 point increment. We randomly sampled 50 samples (LibriTTS 25, VCTK 25) from the evaluation set for the subjective evaluations.[2] We crowd-sourced raters on Amazon Mechanical Turk with a requirement that they live in North America. The total numbers of raters are 165 and 105 for MOS and SMOS, respectively. We also measured the confidence interval (CI) with 95% for all subjective evaluation results.

We also conducted two objective evaluations for speech intelligibility and speaker similarity. The entire evaluation set (500 samples) described in Section 4.1 is used for objective evaluations. We employ an ASR model for the speech intelligibility test to transcribe the generated speech and calculate the character error rate (CER). The ASR model is a CTC-based HuBERT (Hsu et al., 2021) pre-trained on Librilight (Kahn et al., 2020) and fine-tuned on the LibriSpeech dataset. A lower value indicates higher intelligibility for the synthesized speech.

To evaluate the speaker similarity between the ground truth audio and the synthesized audio, we computed the speaker encoder cosine similarity (SECS). We used the state-of-the-art verification

---

[2]Demo: https://osgfqwqbsfcr.github.io/elf-demo/

Table 1: Comparison of evaluation results with previous works.

| Model | Seen/Unseen | SMOS (CI) | MOS (CI) | CER | SECS |
|---|---|---|---|---|---|
| Ground Truth | - | - | 3.95 ($\pm$0.09) | 1.29 | 1.000 |
| VITS multi-speaker | Seen | 3.73 ($\pm$0.09) | 3.95 ($\pm$0.09) | 1.80 | 0.919 |
| YourTTS | Unseen | 3.47 ($\pm$0.11) | 3.88 ($\pm$0.10) | 5.42 | 0.817 |
| ELF (# of audio = 1) | Unseen | 3.62 ($\pm$0.10) | 3.93 ($\pm$0.10) | 2.35 | 0.848 |
| ELF (# of audio = 20) | Unseen | 3.76 ($\pm$0.10) | 3.96 ($\pm$0.09) | 1.26 | 0.879 |
| ELF (all audio) | Unseen | **3.85** ($\pm$0.09) | **3.98** ($\pm$0.09) | 1.49 | 0.888 |

model, WavLM-TDNN (Chen et al., 2022), to extract the speaker vectors. The SECS score is in the range of [-1, 1], with higher values indicating higher similarity.

Table 1 presents the results. Our model significantly outperforms multi-speaker VITS in the SMOS evaluation and shows equivalent performance in the MOS evaluations. Considering that the test speakers were used in training VITS, but not in training our model, these are remarkable results. As shown in the results of CER, our model (# of audio=20 and all audio) achieved lower CER results than the multi-speaker VITS model, exhibiting that our method enables stable speech synthesis with high intelligibility. Our model with one audio condition (# of audio=1) outperforms YourTTS in all evaluations, demonstrating that our method is also substantially effective in the zero-shot scenario.

## 4.2 SPEAKER BLENDING

We conducted subjective and objective evaluations to confirm the quality of the synthesized speech with new artificial speakers using the speaker blending method described in Section 2.4. Table 2 presents MOS and CER with one speaker and generated speakers with a blend of two, four, and eight existing speakers, respectively. The MOS evaluations demonstrate that high-quality speech comparable with the ground truth can be synthesized with new artificial speakers. In addition, the CER results of the synthesized speech with artificial speakers are lower than the ground truth and single speaker. It confirms that our blending method enables synthesizing speech with high intelligibility and stability.

In order to obtain desired speech characteristics when creating a new speaker with existing speakers, controllability to adjust the expression ratio of the original speakers' speech characteristics is needed. Thus, one of the crucial performance factors is how the similarity between speakers changes according to a given proportion. To verify it, we measured and compared SECS with various speaker proportions. Table 4 shows the results of measuring 500 samples generated by a new speaker at the ratios of 2:8, 5:5, and 8:2 for the two speakers (LibriTTS 1580, 1089), respectively. For YourTTS, the blended output at a 5:5 ratio showed a pronounced bias towards speaker B. It shows that even though the point moved in the latent space according to the combination of two vectors at a 5:5 ratio, the speaker similarity is not related to the movement of the point. In other words, speaker similarity is not related to the distances between points. On the other hand, our method demonstrates a considerably uniform change in speaker similarity as the proportion changes, showing even more uniform similarity changes compared to the multi-speaker VITS model. It also implies that the learned latent space of our method is continuous and evenly distributed, allowing for consistent changes in similarity.

Table 2: Evaluation results on the number of blending speakers. (N speaker: N is number of blending speakers)

| Model | MOS (CI) | CER |
|---|---|---|
| Ground Truth | 3.96 (0.09) | 1.29 |
| ELF (1 speaker) | 3.89 (0.10) | 1.19 |
| ELF (2 speakers) | 3.83 (0.10) | 1.01 |
| ELF (4 speakers) | 3.86 (0.09) | 1.07 |
| ELF (8 speakers) | 3.96 (0.09) | 1.03 |

Table 3: Comparison of cross-lingual speech synthesis.

| Model | MOS (CI) | CER | SECS |
|---|---|---|---|
| Ground Truth | 3.99 (0.10) | 1.29 | - |
| ELF | **3.79** (0.12) | 2.84 | 0.862 |
| YourTTS | 3.29 (0.13) | 7.91 | 0.600 |

Table 4: Comparison of similarity changes according to the blending ratio of two speakers. (A:B is the ratio of speaker A and speaker B)

| Model | ELF (Unseen) | | VITS (Seen) | | YourTTS (Unseen) | |
|---|---|---|---|---|---|---|
| | SECS(A) | SECS(B) | SECS(A) | SECS(B) | SECS(A) | SECS(B) |
| speaker A | 1.000 | 0.506 | 1.000 | 0.554 | 1.000 | 0.510 |
| speaker B | 0.506 | 1.000 | 0.554 | 1.000 | 0.510 | 1.000 |
| A:B = 8:2 | 0.943 | 0.597 | 0.919 | 0.659 | 0.950 | 0.546 |
| A:B = 5:5 | 0.742 | 0.800 | 0.760 | 0.824 | 0.651 | 0.909 |
| A:B = 2:8 | 0.568 | 0.936 | 0.609 | 0.932 | 0.533 | 0.966 |

### 4.3 CROSS-LINGUAL TEXT-TO-SPEECH

We measured MOS, CER, and SECS to confirm that our method can synthesize speech similar to an original speaker in a cross-lingual manner with high naturalness and intelligibility. We generate the speech feature codebook with 200 audio clips from the Korean dataset (Park, 2018). We synthesized 500 samples with all texts in the evaluation set using the same TTS model in Section 4.1. As shown in Table 3, the differences between our method and the ground truth in CER and MOS are small, and SECS is comparable with the English speaker results in Table 1. It demonstrates that our method can synthesize high-quality speech similar to the target speaker in the cross-lingual manner. These are remarkable results, considering that English and Korean are fundamentally different languages. We additionally compared with YourTTS, which has an ability for cross-lingual speech synthesis, with randomly selected reference audio for each text. The results show that our method outperforms YourTTS in the cross-lingual manner.

### 4.4 SPEECH FEATURE-TO-SPEECH WITH TEXT CONDITION

To verify the proposed method described in Section 2.5, MOS, CER, and SECS evaluations were conducted. Table 5 shows that MOS and CER differences between our method and ground truth are only 0.05 and 0.07, respectively, and SECS is comparable with the results in Table 1. It demonstrates that high-quality speech can be synthesized solely with the speech features in the codebook, and the codebook is sufficiently informative to reconstruct the target speaker's speech completely. It also implies that our method can be generally used in various tasks where information to combine the speech features in the codebook is available.

Table 5: Comparison of evaluation results on speech feature-to-speech

| Model | MOS (CI) | CER | SECS |
|---|---|---|---|
| Ground Truth | 3.96 (0.09) | 1.29 | - |
| Speech Feature-to-Speech | 3.91 (0.10) | 1.36 | 0.881 |

## 5 CONCLUSION

In this work, we present ELF, a novel method to encode speakers' overall speech features and express speech characteristics of the speakers in high similarity without additional training on the speakers' dataset. In comparison to the best-performing models, ELF performs superior to the multi-speaker model and outperforms the zero-shot model by significant margins. Furthermore, it shows remarkable performance in generating new artificial speakers with the blending method. Also, it presents the ability to synthesize speech with high quality in the cross-lingual manner.

We showed that ELF enables encoding speech features that are sufficiently informative to reconstruct the original speaker's speech completely, and it also enables expressing the speech characteristics with the combining method; it can be used for various tasks as a general methodology to encode and reconstruct speakers' characteristics without training.

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

## A  APPENDIX

### A.1  DISCUSSION ABOUT THE LATENT SPACE OF SPEAKER VERIFICATION TASKS AND OURS

The latent representation required for the speech synthesis task must contain the information needed to reconstruct the speech of a target speaker. However, in the speaker verification task, the latent representation does not require any information to reconstruct the speaker; only information that can distinguish each speaker is required, and the distribution does not matter in any shape. Suppose we divide audio data from a speaker into two groups and train a speaker verification model to classify each group as a different speaker. In that case, the model will try to classify the two groups by capturing any feature in the audio to reduce the loss. It will lead to a latent representation learned with little to do with speaker similarity and reconstruction. The distribution obtained from high-performance speaker verification tasks can not be continuous and evenly distributed. If this distribution is continuous and even, it becomes difficult to form a decision boundary, which means that the performance of the speaker verification model is poor. (Figure 4a shows the distribution of the vectors obtained from the speaker encoder (Casanova et al., 2022) of a speaker verification model.) Therefore, sampling a speaker vector from discontinuous areas that the synthesis model has never or rarely learned is often possible. Moreover, since the similarity of speakers is not related to the distance between the speaker vectors, the synthesis model tends to learn each vector as an independent condition rather than a space. Therefore, if there is a speaker with a very similar vector among the learned speakers, it is successfully synthesized, but if not, it fails to synthesize correct speech, or it often synthesizes speech with definitely different characteristics. In addition, when blending speakers, the sampled point is more likely to be in the not-learned position, which also increases the possibility that synthesis with an artificial speaker will fail. The problems of the previous methods are frequently observed and can be confirmed in the comparison results presented in our paper.

We use VAE to encode speech features. In VAE, the aggregated posterior can not perfectly fit the prior, and the training of a model is not ideal; thus, it is difficult to see that the learned space follows the Gaussian distribution perfectly, but theoretically and empirically, it follows a distribution close to Gaussian. Figure 4b shows the distribution of the entire speakers' codebooks of our method. Since it is a dimensionality-reduced high-dimensional distribution, it can not demonstrate the exact distribution, but it is noticed that it is a continuous and dense distribution.

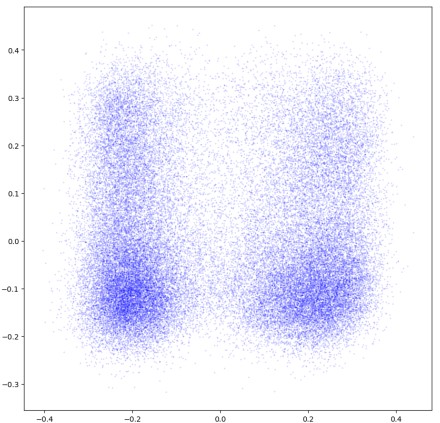 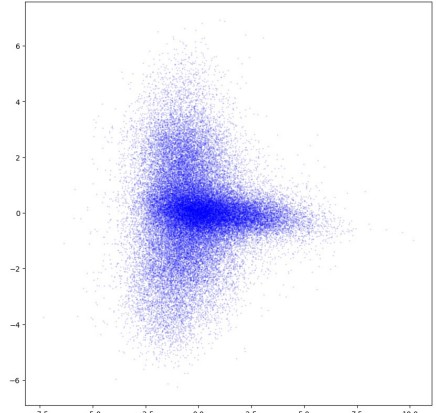

(a) Distribution of the vectors obtained from the speaker encoder of a speaker verification model.

(b) Distribution of the entire speakers' codebooks of our method.

Figure 4: Visualization of two distributions. PCA is used for dimensionality reduction.

## A.2 NOISE ROBUSTNESS

Because our method encodes speech features from raw waveforms, evaluating noise robustness is valuable in terms of whether our method can be a practical solution. Therefore, we conducted experiments by introducing noise to clean audio to evaluate noise robustness. We sampled a test speaker not included in training and added randomly sampled noises from the TAU Urban Acoustic Scenes 2021 Mobile Evaluation dataset with a signal-to-noise ratio (SNR) of 15 dB. Then, we encoded them using SFEN, generated a codebook, and synthesized TTS samples. We measured CER and SECS; the results are shown in Table 6. The evaluation results show that our method maintains high performance even with significant noise.

Table 6: Evaluation results for noise robustness

| Model | CER | SECS |
|---|---|---|
| Ground Truth | 1.29 | - |
| ELF with clean audio | 1.49 | 0.888 |
| ELF with noisy audio | 2.07 | 0.865 |

## A.3 VISUALIZATION OF THE SPEAKER VECTORS FROM SYNTHESIZED SPEECHES

To confirm that the synthesized speeches with our method present distinguishable speech characteristics from each other, we visualized the speaker vectors of the synthesized speeches of unseen speakers. We used the speaker encoder in the previous work (Casanova et al., 2022) to obtain the speaker vectors and employed t-SNE for visualization. As shown in Figure 5, each speaker's vectors are closely clustered, and the speakers are distinguished clearly.

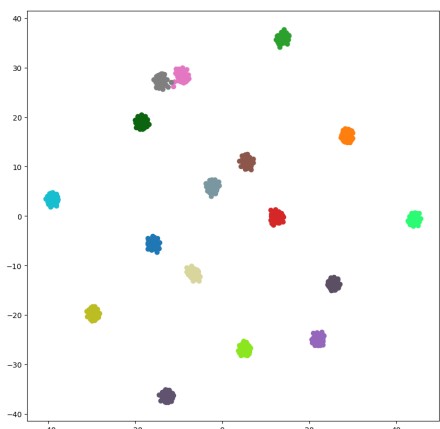

Figure 5: Visualization of the speaker vectors from synthesized speeches

## A.4 PRIOR ENCODER OF SPEECH FEATURE-TO-SPEECH

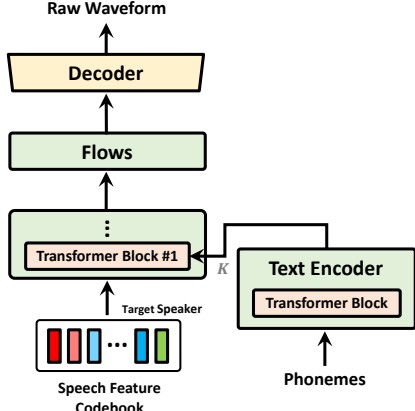

Figure 6: Prior encoder (green blocks) of speech feature-to-speech model.

## A.5 DETAILS OF SFEN TRAINING

The networks were trained using the AdamW optimizer (Loshchilov & Hutter, 2019) with $\beta_1 = 0.8$, $\beta_2 = 0.99$, and weight decay $\lambda = 0.01$. The learning rate decay was scheduled by a $0.999$ factor in every epoch with an initial learning rate of $2 \times 10^4$. 8 NVIDIA V100 GPUs were used to train the model. The batch size was set to 32 per GPU, and the model was trained up to 800k steps.

## A.6 DETAILS OF TTS TRAINING

The networks were trained using the AdamW optimizer (Loshchilov & Hutter, 2019) with $\beta_1 = 0.8$, $\beta_2 = 0.99$, and weight decay $\lambda = 0.01$. The learning rate decay was scheduled by a factor of $0.998$ in every epoch, with an initial learning rate of $2 \times 10^4$. 8 NVIDIA V100 GPUs were used, and the batch size was set to 32 per GPU and trained up to 500k steps. Because we use similar generator and discriminator architecture and data in both SFEN and the TTS model, we initialized the last three of the four residual blocks in the generator and the discriminators of the TTS model with the parameters of SFEN. It speeds up the training; therefore, the model performs better than random initialization at the same training step.

