# OpenReview forum: "Encoding Speaker-Specific Latent Speech Feature for Speech Synthesis"
_ICLR.cc/2024/Conference — Submitted to ICLR 2024_

### Official Review · Reviewer_reHV · 2023-10-25

**Soundness:** 2 fair
**Presentation:** 2 fair
**Contribution:** 3 good
**Rating:** 5
**Confidence:** 4

**Summary:**

In the paper, the authors proposed a new method for modeling speaker timbre. Specifically, to encode the speaker characterize, a VAE-like model is trained to encode speech to μ and σ. And then the μ is used to cluster to several cookbooks that represent the speaker information. Through the attention alignment between text representation and codebook, the text and speaker representation are fused into synthesis speech. The experiments shows the propose method performs superior to previous multi-speaker model and outperforms a zero-shot model.

**Strengths:**

1. A new speaker modeling method is proposed for zero-shot generation.
2. The demo provided by the authors seems convincing.
3. The experiment results show the performance of the proposed method.

**Weaknesses:**

1. The relations between previous works and this paper are not well presented in Section I.  "The zero-shot method obtains a speaker vector from a short reference audio, and the timbre and prosody expressed in the given reference audio are aligned with its content. In other words, only a small portion of the speech characteristics that the speaker can express can be obtained from the reference audio." In recent studies, many methods try to capture more speaker timbre from reference speech. These methods share a similar process, which is to encode speech to serval vectors and then use attention to fuse the text and speaker timbre. For example, in TTS, https://www.isca-speech.org/archive/pdfs/interspeech_2022/zhou22d_interspeech.pdf, NANSY++, RetrieverTTS.
2. The proposed method is not presented very clearly. For example:
2.1 why μ is chosen for clustering
2.2 how to perform the clustering
2.3 how many codebooks are used for representing a speech.  Does the number of codebooks have a relation to the number of cluster centers?
2.4 how to obtain the codebook and what is the number of cookbooks in the codebook set?
3. As mentioned in W.1,  many previous zero-shot TTS methods also represent reference speech to a set of vectors and fuse them with text via attention. Though I can understand the proposed system and comparison system all use VITS as the backbone for fair comparison, I think it better to compare your speaker modeling method with the others for a better understanding of the superiority, since the ELF is the main claimed contribution.

**Questions:**

/

---

> ### Author Response · Authors · 2023-11-19
>
> We thank the reviewer for the valuable comments.
>
> **W1** \
> We agree that supplementing the manuscript with the mentioned related works is valuable. \
> Our core goal is to naturally express a speaker's overall speech characteristics depending on input content, like a single-speaker model or multi-speaker model. Examples of this are available in the second section of our demo page (newly added in the middle of the review period). \
> Previous works [1,2,3] used multiple feature vectors but still focused on following short reference audio. Take the following two sentences as examples: "I'm so happy!" and "I'm so sad.". The speaker's psychological state uttering the two sentences will be greatly different, and it will be revealed as speech characteristics. If the sentence "I'm so sad." is synthesized using the methods of the previous works and a recorded speech audio of "I'm so happy!" is given as the reference audio, the synthesized speech will be unnatural and show quite different speech characteristics from the actual speech when the target speaker utters "I'm so sad.". it will be synthesized with a bright timbre and prosody contrasting with typical human speech. \
> Since previous methods do not condition a speaker's overall speech characteristics in training the TTS models, they do not allow the TTS model to learn how to select and fuse speakers' speech characteristics to express similarly according to current content. Therefore, even if fairly long reference audio or concatenated feature vectors are conditioned to the TTS models, it is not easy to expect to effectively express the speech characteristics of the target speaker depending on a given content. Additionally, this method can require a large amount of computation, and the length of features varies for each speaker, making it difficult to predict the required amount of computation and memory size. \
> Our method is significantly different from the previous works in that it allows speakers' overall speech characteristics to be expressed while using feature vectors of a constant size regardless of the length of the given audio. We have summarized and supplemented this content in the manuscript.
>
> **W2** \
> Since the posterior fit on Gaussian distribution, the mean value represents the highest probability point of the feature distribution. Therefore, using the mean value(μ) rather than using other points is a good representative point for expressing the feature distribution.
> We described our method for clustering and constructing the codebook in Section 3.2.
> In our work, the codebook consists of centroids of 512 clusters, and k-means++ is used to cluster.
>
> **W3** \
> Although it can be valuable to make comparisons in the manner mentioned by the reviewer, the lack of sufficient implementation details to reproduce the performance claimed by previous works makes such comparisons difficult. We believe that our comparison with state-of-the-art works using the same backbone has sufficiently shown the effectiveness of our method. Additionally, to the best of our knowledge, our work represents a significant advance in that it is the first to show superior performance over the multi-speaker model trained on the target speakers dataset in a non-training approach.
>
>
> [1] Zhou, Yixuan, et al. "Content-dependent fine-grained speaker embedding for zero-shot speaker adaptation in text-to-speech synthesis." arXiv preprint arXiv:2204.00990 (2022).
>
> [2] Yin, Dacheng, et al. "RetrieverTTS: Modeling decomposed factors for text-based speech insertion." arXiv preprint arXiv:2206.13865 (2022).
>
> [3] Choi, Hyeong-Seok, et al. "NANSY++: Unified voice synthesis with neural analysis and synthesis." arXiv preprint arXiv:2211.09407 (2022).

---

> ### Author Response · Authors · 2023-11-22
>
> We greatly appreciate the reviewer's time and interest in reviewing our work. We hope that our response has addressed the reviewer's concerns and questions. Please let us know if the reviewer has concerns that have not been properly addressed. We will do our best to answer them.

---

> > ### Comment · Reviewer_reHV · 2023-11-22
> >
> > Thanks for your response. But I still have some questions.
> >
> > W2. What is the impact of different codebook sizes (512 clusters in paper) on the model's performance?
> >
> > W3. I fully understand the difficulty of reproducing these systems. Since the proposed method actually has the ability to achieve speaker modeling from single and several utterances and obtain a variable-length speaker representation,  only selecting Yourtts (single speaker vector, different kind speaker modeling method) for zero-shot comparison seems insufficient for proving the SOTA performance of the proposed speaker modeling method.

---

> ### Author Response · Authors · 2023-11-22
>
> We sincerely thank the reviewer for the valuable comments.
>
> **W2** \
> In the early stage of our work, we experimented with 128, 256, and 512 clusters and observed that performance improved slightly as the number of clusters increased. Considering that the performance difference between the 256 clusters setting and the 512 clusters setting is less than 3% in the evaluation of SECS and that the time required for clustering increases significantly as the number of clusters increases, we chose 512 as the number of clusters and conducted studies based on this.
>
> **W3** \
> We acknowledge that our previous response was insufficient. First, we do not claim in any part of our paper that our work is state-of-the-art. Recently, various new methods have been proposed, and we believe that it is difficult to accept a claim that any work is state-of-the-art unless it is compared fairly with other related works that are reproduced as the performance presented in its paper. \
> In that, we also do not claim that our work is state-of-the-art. What we explicitly claim is that our work outperforms the trained multi-speaker model, considered state-of-the-art in the community, in a non-training manner. \
> To the best of our knowledge, there is no work that has demonstrated these comparison results and performance.
>
>
> We hope that our response adequately addresses the reviewer's concerns. If you have any additional comments, please let me know. We will do our best to address it.

---

### Official Review · Reviewer_8YZU · 2023-10-28

**Soundness:** 2 fair
**Presentation:** 3 good
**Contribution:** 1 poor
**Rating:** 3
**Confidence:** 3

**Summary:**

The work introduces a method for speaker modeling targeting speech synthesis, capturing their characteristics without specific training on each speaker's dataset. This approach outperforms existing methods and can generate artificial speakers effectively. The encoded features are claimed as informative enough to fully reconstruct an original speaker's speech, making it versatile for various applications.

**Strengths:**

1. The paper addresses

**Weaknesses:**

1. The connection between the method itself and earlier works are not very clear. The author should have a short review of the related literature.
2. The methods acquired themselves are sometimes a bit abrupt and lacks motivation.
3. The prototype of the model does not show very notable improvements in metrics other than MOS scores for audio. Speaker blending should be applied to reach better performance.
4. The study also lacks comparison with other fixed/earlier speech encoders or speaker encoders.

**Questions:**

1. Possibly because of the first point in weaknesses, in Section 2.2, I do not see strong motivation of acquiring autoencoders, despite there are multiple speaker encoding methods available (e.g. speaker encoders; speech factorization methods). Could you please clarify the motivation?
2. Why speaker blending is needed to reach better CER than the ground truth? And is there any alternative to compensate?

**Details Of Ethics Concerns:**

There is no ethical concern for this study from the reviewer's perspective.

---

> ### Author Response · Authors · 2023-11-19
>
> We thank the reviewer for the valuable comments.
>
> **W1, W2 and Q1** \
> Our method aims to express the overall speech characteristics of a target speaker rather than only the speech characteristics presented in a short reference audio. Examples of it are available in the second section of our demo page (newly added in the middle of the review period). \
> In order to enable the TTS model to synthesize speech according to the overall speech characteristics of an unseen speaker and a given content, it is necessary to show the speaker's overall speech characteristics to the TTS model in training and to design the TTS model to combine the given speech characteristics according to the given content. It cannot be resolved with previous speaker encoders that take short reference audio as input and the TTS models [1,2,3] inputted with the output of the speaker encoders. \
> To address it, our method consists of two stages: 1) encoding speakers' speech characteristics and storing the encoded features with discretization; 2) conditioning them to synthesize a target speaker's speech with a given content. \
> Considering that the goal of our work is to synthesize a target speaker's speech similarly, the representation of speech characteristics for this should be sufficiently informative to reconstruct the target speaker's speech completely. An autoencoder serves the above purpose adequately by converting the input into a latent representation and reconstructing it back to the original input. We have summarized and supplemented this in the manuscript.
>
> **W3** \
> As shown in Table 1, our method is superior to the trained multi-speaker model VITS in the SMOS evaluation. It shows a more significant SMOS difference than the previous zero-shot approach model YourTTS. These differences are significantly larger than the differences in MOS evaluation results. To the best of our knowledge, our work is the first that shows superior results to the best-performing multi-speaker model trained on a target speaker's dataset with a non-training method. To enable intuitive confirmation in the table, we marked "Seen" for models trained with a target speaker's data and "Unseen" for models that were not trained on the dataset.
>
> **W4** \
> We compared our method with YourTTS, which uses a different speaker encoder. This work has been cited as a crucial comparative work in recent works [4,5,6] and was recently recognized as state-of-the-art. We increase the fairness of comparison by using the same backbone. Although comparisons with a wider range of other works would also be valuable, we believe that we have sufficiently demonstrated the effectiveness and contribution of our method by showing its superiority over the model VITS trained with a target speaker's dataset.
>
> **Q2** \
> Since the ASR model is trained with data from numerous speakers, it learns pronunciation and speech characteristics commonly observed across many speakers at a higher frequency. If the speaker space we trained is fitted according to the numerous speakers' real distribution, blending multiple speakers is equivalent to approaching speech characteristics with a higher probability of occurrence. Therefore, theoretically, it is natural that blending more speakers results in better CER. It is also evidence that the speaker space we trained represents the real distribution of speakers well.
>
> [1] Zhou, Yixuan, et al. "Content-dependent fine-grained speaker embedding for zero-shot speaker adaptation in text-to-speech synthesis." arXiv preprint arXiv:2204.00990 (2022).
>
> [2] Yin, Dacheng, et al. "RetrieverTTS: Modeling decomposed factors for text-based speech insertion." arXiv preprint arXiv:2206.13865 (2022).
>
> [3] Choi, Hyeong-Seok, et al. "NANSY++: Unified voice synthesis with neural analysis and synthesis." arXiv preprint arXiv:2211.09407 (2022).
>
> [4] Wang, Chengyi, et al. "Neural codec language models are zero-shot text to speech synthesizers." arXiv preprint arXiv:2301.02111 (2023).
>
> [5] Shen, Kai, et al. "Naturalspeech 2: Latent diffusion models are natural and zero-shot speech and singing synthesizers." arXiv preprint arXiv:2304.09116 (2023).
>
> [6] Le, Matthew, et al. "Voicebox: Text-guided multilingual universal speech generation at scale." arXiv preprint arXiv:2306.15687 (2023).

---

> > ### Comment · Reviewer_8YZU · 2023-11-21
> >
> > Thanks to the authors for replying.
> >
> > W1: Thanks for such fruitful response. Then I wonder if you would like to have the response into the paper?
> >
> > Q1: Then could you please brief the potential benefit of using latent representations, instead of other speaker representations?

---

> > > ### Author Response · Authors · 2023-11-21
> > >
> > > We greatly appreciate the reviewer's response.
> > >
> > > **W1** \
> > > We will refine the content of the previous response and apply it to the final version. Based on the reviewer's comments, we will explain our work clearly so readers can better understand it.
> > >
> > > **Q1** \
> > > The speaker representation presented in the previous works [1,2,3] is obtained from a speaker encoder trained with a speaker verification task. The objectives of speaker verification tasks are designed to distance each speaker on its speaker space to distinguish them well. This method forms a discontinuous sparse distribution, as shown in Figure 4 in Appendix A.1. Therefore, when sampling points (representations) for an unseen speaker from the learned speaker space, the possibility to sample points from locations that the TTS model has not learned or sparsely learned is high. \
> > > Additionally, since the objective of speaker verification is to distance each speaker regardless of their similarity, speaker similarity and the distance between points (representations) have little to do, and the TTS model tends to learn each speaker's vector as an individual value rather than a space. Lastly, using a linear combination of multiple points (representations) in this space to utilize the speaker's overall speech characteristics makes the problem more severe and is theoretically implausible. \
> > > The previous work [4] utilizes a speaker encoder that is jointly trained with the TTS model. Although this method does not force speakers to be distanced, it still has the similar problems described earlier because it does not regularize the speaker space to be continuous and dense. Please refer to the "Optimizing purely for reconstruction loss" figure on the following web page. (https://towardsdatascience.com/intuitively-understanding-variational-autoencoders-1bfe67eb5daf) \
> > > Therefore, this method will also likely sample points (representations) from unlearned or sparsely learned locations for unseen speakers. Additionally, there is no theoretical basis to claim that a linear combination of multiple points (representations) sampled in the space can be an actual speech feature located somewhere in the middle of those points. \
> > > We use VAE to ensure that the distribution of all speaker features is close to the unit Gaussian. More details are in Appendix A.1 and Figure 4. This theoretical basis makes it natural that we extract representative points among the countless features with simple distance-based clustering and linearly combine them to reconstruct the speech characteristics of target speakers. In addition, in Appendix A of the previous work [5], we can see that the reconstructed results continuously change according to the movement of data points in the learned latent space. If our proposed speaker feature space is well learned, as in the previous work [5], when the sampled speech features move, they will change to adjacent features according to the distance and direction. It can be confirmed through our speaker blending experiments. \
> > > These theoretical backgrounds and empirical results show that the proposed learned space is continuously and densely formed according to speakers' similarities. It significantly improves the similarity and robustness of the synthesis for unseen speakers by enabling avoidance of sampling from sparsely learned positions and allowing the TTS model to learn the conditioned features of speakers as a space. The superiority of the robustness can be confirmed through the CER results we presented. Finally, by adopting a known distribution, it is possible to discretize the space into a few sampled representative points and effectively restore original points from the continuous space formed by the representative points. This method can be used in various tasks that require encoded speech features. It can also be confirmed in Sections 3.4 and 4.4 in our paper.
> > >
> > >
> > > [1] Jia, Ye, et al. "Transfer learning from speaker verification to multispeaker text-to-speech synthesis." Advances in neural information processing systems 31 (2018).
> > >
> > > [2] Cooper, Erica, et al. "Zero-shot multi-speaker text-to-speech with state-of-the-art neural speaker embeddings." ICASSP 2020-2020 IEEE International Conference on Acoustics, Speech and Signal Processing (ICASSP). IEEE, 2020.
> > >
> > > [3] Casanova, Edresson, et al. "Yourtts: Towards zero-shot multi-speaker tts and zero-shot voice conversion for everyone." International Conference on Machine Learning. PMLR, 2022.
> > >
> > > [4] Zhou, Yixuan, et al. "Content-dependent fine-grained speaker embedding for zero-shot speaker adaptation in text-to-speech synthesis." arXiv preprint arXiv:2204.00990 (2022).
> > >
> > > [5] Kingma, Diederik P., and Max Welling. "Auto-encoding variational bayes." arXiv preprint arXiv:1312.6114 (2013).

---

### Official Review · Reviewer_WfZD · 2023-10-29

**Soundness:** 3 good
**Presentation:** 2 fair
**Contribution:** 2 fair
**Rating:** 5
**Confidence:** 4

**Summary:**

In this work authors obtain a zero-shot speech cloning method via clustering. VAE type model is used to model the utterance latent space, which is then clustered and the key idea seems to be that each speaker falls into multiple clusters. Final speaker representation is the mix of these cluster centroids. When the query utterance is fed into the system it follows the same path as in training. So speakers not seen in training can be utilized cloned achieving zero-short method.

**Strengths:**

- Neat key idea, where speakers fall into multiple clusters. This achieves modeling diversity of speakers voice characteristics.
- Good empirical results.

**Weaknesses:**

- Final SFEN objective (Eqs. 2-3) comes out of thin air, it would be good to somehow try to explain theoretically how this can be derived, noting that VAE loss is derived from trying to model log p(x).
- Continuing the above critizism, elements in the loss are by necessity weighted somehow. How you decide the proper weighting?
- Plotting cosine scores as objective speajer recognition is ok, but proper speaker recognition results are needed. As in cosine score you only compare against targer speaker and totally miss the confusion with the non-target speaker. So presenting EER and minDCF values with the accomanying DET plots are necessary.
- Paper neeeds more thorough language editing but this can be performed in rebuttal stage. Abstract was badly written, but some other parts of the paper are quite ok.

**Questions:**

- I see that MT was used to obtain MOS and SMOS scores. How did you clean MT results as those can be sometimes extremely noisy. MT workers sometimes use scripting to speed up the work and so those those workers should be removed from the results.
- How did you measure CI for MOS and SMOS?
- Were all samples finally voceded using the same voceder before objective speaker recognition? If not, then you can easily add (inaudible) vocoder artifacts that your TDNN model can then pick up. Vocode all samples, even ground truth ones, using the same vocoder.
- Why in Table 1, proposed CER is lower in #20 than in all audio?

---

> ### Author Response · Authors · 2023-11-19
>
> We thank the reviewer for the valuable comments.
>
> **W1** \
> Our work presents a method combining a variational autoencoder(VAE) and adversarial learning. It significantly improves the reconstruction performance of SFEN. Therefore, the objective we present consists of terms for optimizing generative adversarial networks (GAN) and VAE. In the previous work [1, 2], the authors proved that these objectives are to model a real data distribution. Other widely cited works [3, 4] showed that modeling a real data distribution from an unknown input distribution is possible with the same theoretical background as the first work [1]. Thus, it is also possible to model the real data distribution by applying the adversarial learning method with the output distribution of the SFEN encoder. In conclusion, both the VAE and adversarial learning terms are aligned to model a real data distribution. Additionally, we follow LSGAN [5] as the objective of adversarial learning, which presents a solid theoretical background that allows GAN to model real data distribution better.
>
> **W2** \
> We followed the coefficient 45 for l1 reconstruction loss presented in the previous work [4] and did not use weight for other terms. We agree that details were omitted from the manuscript and have revised it.
>
> **W3** \
> We believe that the SMOS and SECS results have sufficiently demonstrated the effectiveness of our method. Also, subjective evaluation still carries more weight than objective evaluation in evaluating synthesized speech. The speaker vector plot we presented in the appendix is auxiliary data that demonstrates the superiority of our method. Referring to the related previous works [6,7,8,9,10], we can confirm that the metrics mentioned by the reviewer are unnecessary. In addition, as described in the manuscript, 21 test speakers are in our experimental settings. Measuring speaker verification performance with such few speakers does not provide valid evidence.
>
> **W4** \
> We have made minor corrections to the abstract. We will thoroughly review it again to enhance readability for readers.
>
> **Q1** \
> We agree that removing the mentioned noisy data is important when utilizing the crowd-sourced MOS method via Amazon Mturk. We removed noisy data by adding multiple evaluation samples that were not related to the comparisons, considering workers who gave clearly incorrect evaluations as random workers, and excluding all evaluation results from those workers.
>
> **Q2** \
> The mean and CI were calculated by aggregating all scores for each comparative method.
> We followed the commonly used method of measuring 95% confidence intervals in statistics. Detailed explanations are in the materials below.
> http://www.stat.yale.edu/Courses/1997-98/101/confint.htm
>
> **Q3** \
> Our method and the comparative methods do not use vocoders. Entire modules in the models are jointly trained; therefore, there are no separately operating vocoders.
>
> **Q4** \
> The difference between the two CER results is minimal. This degree of difference can be caused due to the randomness of model training. We see that they are almost identical results, and it is difficult to state that the robustness of the two methods is different.
>
>
> [1] Goodfellow, Ian, et al. "Generative adversarial nets." Advances in neural information processing systems 27 (2014).
>
> [2] Kingma, Diederik P., and Max Welling. "Auto-encoding variational bayes." arXiv preprint arXiv:1312.6114 (2013).
>
> [3] Isola, Phillip, et al. "Image-to-image translation with conditional adversarial networks." Proceedings of the IEEE conference on computer vision and pattern recognition. 2017.
>
> [4] Kong, Jungil, Jaehyeon Kim, and Jaekyoung Bae. "Hifi-gan: Generative adversarial networks for efficient and high fidelity speech synthesis." Advances in Neural Information Processing Systems 33 (2020): 17022-17033.
>
> [5] Mao, Xudong, et al. "Least squares generative adversarial networks." Proceedings of the IEEE international conference on computer vision. 2017.
>
> [6] Casanova, Edresson, et al. "Yourtts: Towards zero-shot multi-speaker tts and zero-shot voice conversion for everyone." International Conference on Machine Learning. PMLR, 2022.
>
> [7] Hsu, Wei-Ning, et al. "Hierarchical Generative Modeling for Controllable Speech Synthesis." International Conference on Learning Representations. 2018.
>
> [8] Shen, Kai, et al. "Naturalspeech 2: Latent diffusion models are natural and zero-shot speech and singing synthesizers." arXiv preprint arXiv:2304.09116 (2023).
>
> [9] Wang, Chengyi, et al. "Neural codec language models are zero-shot text to speech synthesizers." arXiv preprint arXiv:2301.02111 (2023).
>
> [10] Le, Matthew, et al. "Voicebox: Text-guided multilingual universal speech generation at scale." arXiv preprint arXiv:2306.15687 (2023).

---

> ### Author Response · Authors · 2023-11-22
>
> We greatly appreciate the reviewer's time and interest in reviewing our work. We hope that our response has addressed the reviewer's concerns and questions. Please let us know if the reviewer has concerns that have not been properly addressed. We will do our best to answer them.

---

> > ### Comment · Reviewer_WfZD · 2023-11-22
> > **Reviewer comment after rebuttal**
> >
> > Partially I agree with authors in their rebuttal. One clear problem is in experimental evaluation, where authors, for some reason, did not want to use speaker veritication -based objective evaluation metrics. And consequently, for Q3, I find that authors did not probably understand my point. As if you compare clean audio (without vocoding) and some vocoded audio, such as the proposed method. Then by necessity there are some confounders present that only come from the vocoder artefacts. Authros should think how they can vocode also the clean speech and thus obtain comparable audio files.
> >
> > Theoretical basis of the loss is still missing even after rebuttal. In some sense it is ok to have purely empirical work. But some theoretical justification for the loss would definitely be beneficial.
> >
> > I will keep my reviewer score as is.

---

> ### Author Response · Authors · 2023-11-22
>
> > Partially I agree with authors in their rebuttal. One clear problem is in experimental evaluation, where authors, for some reason, did not want to use speaker veritication -based objective evaluation metrics. And consequently, for Q3, I find that authors did not probably understand my point. As if you compare clean audio (without vocoding) and some vocoded audio, such as the proposed method. Then by necessity there are some confounders present that only come from the vocoder artefacts. Authros should think how they can vocode also the clean speech and thus obtain comparable audio files.
>
> Let us explain again what we have presented in the paper.
>
> 1. We have never presented the speaker recognition results that the reviewer mentioned in Q3. \
> Therefore, there is no comparison that distinguishes between clean audio and synthesized audio (the reviewer mentioned "vocoded", but since none of the models in our paper use a vocoder, we'll say "synthesized"). \
> 2. The speaker vector plot we presented in Appendix A.3 are all for synthesized audio, so they are not related to the mentioned vocoder artifact. \
> 3. The WavLM-TDNN model was used to measure SECS, not speaker recognition. Since SECS were measured between ground truth audio and synthesized audio for all models, the evaluation conditions are fair. \
>
> We'd like to ask the reviewer where we should apply a vocoder. \
> The discussion will be more effective if the reviewer specifies it.
>
>
> > Theoretical basis of the loss is still missing even after rebuttal. In some sense it is ok to have purely empirical work. But some theoretical justification for the loss would definitely be beneficial.
>
> The theoretical basis for modeling a real data distribution mentioned by the reviewer is explained in detail in Section 4 in the previous work [1] and Section 3 in the previous work [2]. If the reviewer specifically points out parts that require more theoretical basis for modeling a real data distribution, we will explain those parts.
>
> [1] Goodfellow, Ian, et al. "Generative adversarial nets." Advances in neural information processing systems 27 (2014).
>
> [2] Mao, Xudong, et al. "Least squares generative adversarial networks." Proceedings of the IEEE international conference on computer vision. 2017.
>
> We hope that effective discussions will proceed through specified comments, \
> and we also hope that our work will be reviewed based on a sufficient understanding of our work and the related works.

---

> > ### Comment · Reviewer_WfZD · 2023-11-22
> > **Answer to authors**
> >
> > Whenever you synthesize audio, you are going to use some "vocoder" -type mechanism. Whether you directly sample samples, as in wavenet, you in effect use some vocoding mechanism. And as such this mechanism will cause artefacts. Then by comparing ground truth audio, which I guess is "clean audio", you are in effect measuring also these artefacts. Your SECS _IS_ speaker verification measure, but as I explained before you are measuring only target speaker scores and you _ARE_ missing non-target speaker scores. Measure that will take all of these into account is for example EER and minDCF.
> >
> > About theoretical basis, I am _NOT_ talking about theory of GANs. But I am saying that how you would derive your loss function. What assumptions you would need to be able to obtain the proposed loss. Sometimes you can obtain a partial explanation, like loss forms additive terms X, Y and Z. But then you experimentally add some tunable coefficients there.

---

> ### Author Response · Authors · 2023-11-23
>
> We provide supplementary explanations and the EER and minDCF data requested by the reviewer.
>
> > Whenever you synthesize audio, you are going to use some "vocoder" -type mechanism...
>
> 1. Obtaining high SECS values is advantageous for our work. As mentioned in the previous responses, our comparison is between ground truth audio and synthesized audio. As the reviewer mentioned, if the speaker verification model can easily distinguish between the ground truth audio and synthesized audio due to the influence of artifacts, SECS will be lower. Therefore, the evaluation settings we presented are more stringent than those that eliminate the influence of artifacts.
>
> 2. Nevertheless, since there is a possibility that models with different structures have individual characteristics, we used the same backbone as the comparison models.
>
> 3. In the Wavenet [1] era, TTS system was composed of a two-stage structure using mel-spectrogram as a medium. However, the models we used do not use intermediate features such as mel-spectrograms that can be easily obtained from raw waveforms. Please refer to the illustration below.
>
> ---
> Tacotron2 + Wavenet [1], FastSpeech [3], Glow-TTS [4] and HiFi-GAN [5] :
>
> Text —> [1st stage model]—> mel-spectrogram —> [2nd stage model (vocoder)] —> waveform
>
> ---
> Our work, VITS and YourTTS :
>
> Text —> [model] —> waveform
>
> ---
>
> Therefore, there is no way to compare ground truth audio by vocoding it. If we introduce a vocoder that is unrelated to our task, vocoding and comparing all the audio, the quality degradation makes it difficult to evaluate the actual performance of the model accurately. Also, considering that the output of our model is a complete waveform and is directly used in applications, it is not a reasonable evaluation setting.
>
> 4. We measured EER and minDCF, as requested by the reviewer. We used the same speaker verification model mentioned in our paper. As mentioned in the previous response, since it consists of a much smaller number of test speakers than evaluations in typical speaker verification tasks, the result values are also very small. We hope that the reviewer's concerns will be addressed.
>
> All measurements except ground truth were conducted between ground truth and synthesized audio.
>
> **Ground Truth** \
> EER : 0.099 \
> minDCF : 0.020
>
> **VITS - Seen Speakers** \
> EER : 0.108 \
> minDCF : 0.019
>
> **YouTTS - Unseen Speakers** \
> EER : 0.201 \
> minDCF : 0.045
>
> **ELF (# of audio = 1) - Unseen Speakers** \
> EER : 0.169 \
> minDCF : 0.039
>
> **ELF (# of audio = 20) - Unseen Speakers** \
> EER : 0.158 \
> minDCF : 0.042
>
> **ELF (all audio) - Unseen Speakers** \
> EER : 0.112 \
> minDCF : 0.040
>
> ---
>
> > About theoretical basis, I am NOT talking about theory of GANs....
>
> We will break down the loss equations and explain them.
> Eq (2) in the manuscript is the same as the loss presented in the previous works [6, 5] for training the discriminator.
> Eq (3) consists of a total of 4 terms. The first and second terms are for adversarial learning. These are the same as the terms presented in the previous work [5]. The second term is used in the previous [5] work under the name "feature-matching loss", which is described in detail in Section 2.3 of the previous work [7]. Next, the third term is reconstruction loss, assuming that the data distribution is a Laplace distribution. We applied the coefficient presented in the previous work [5] for this term. The fourth term is the KL divergence of VAE.
>
> The reviewer's specified comments were very helpful. \
> We hope our efforts and responses adequately address the reviewer's concerns.
>
> [1] Oord, Aaron van den, et al. "Wavenet: A generative model for raw audio." arXiv preprint arXiv:1609.03499 (2016).
>
> [2] Shen, Jonathan, et al. "Natural tts synthesis by conditioning wavenet on mel spectrogram predictions." 2018 IEEE international conference on acoustics, speech and signal processing (ICASSP). IEEE, 2018.
>
> [3] Ren, Yi, et al. "Fastspeech: Fast, robust and controllable text to speech." Advances in neural information processing systems 32 (2019).
>
> [4] Kim, Jaehyeon, et al. "Glow-tts: A generative flow for text-to-speech via monotonic alignment search." Advances in Neural Information Processing Systems 33 (2020): 8067-8077.
>
> [5] Kong, Jungil, Jaehyeon Kim, and Jaekyoung Bae. "Hifi-gan: Generative adversarial networks for efficient and high fidelity speech synthesis." Advances in Neural Information Processing Systems 33 (2020): 17022-17033.
>
> [6] Mao, Xudong, et al. "Least squares generative adversarial networks." Proceedings of the IEEE international conference on computer vision. 2017.
>
> [7] Larsen, Anders Boesen Lindbo, et al. "Autoencoding beyond pixels using a learned similarity metric." International conference on machine learning. PMLR, 2016.

---

> > ### Comment · Reviewer_WfZD · 2023-11-23
> > **Reviewers response**
> >
> > Ok thanks a lot for your detailed explanation. I do get the point why now in the modern TTS era vocoding block is impossible to separate and apply to ground truth audio. However, you probably still agree that any mechanism used to generate samples will in effect produce artefacts. I agree that taking those artefacts into account is beyond the present work.
> >
> > Thanks a lot for proviuding EERs and minDCFs those are helpful for the reader. I hope those can be added to the appendix.
> >
> > I can raise my score by one point.

---

### Meta-Review · Area_Chair_2DNJ · 2023-12-02

**Metareview:**

The authors present a new zero-shot speech clodning method via clustering. More specifically, an utterance in the latent space is modeled via VAE and then clustered. A speaker representation is then obtained by mixing centroids. In this way the proposed approach captures speaker's characteristics without specific training. On the one-hand, the proposed idea is neat, and has some merits. On the other end, some aspects still remain unclear, such as the perfomance of the system compared to SOTA - although the authors have attemped a reply, it does not seem fully addressing the raised concerns. Although taking artefacts into account has been claimed to beyond the presented work, it is still an important aspect to be considered.

**Justification For Why Not Higher Score:**

There are a few aspects that still ned to be taken into account despite the merit of the idea.

**Justification For Why Not Lower Score:**

N/A

---

### Decision · Program_Chairs · 2024-01-16

Reject